# The Target-Defining Attributes Can Determine the Effects of Attentional Control Settings in Singleton Search Mode

**DOI:** 10.3390/bs15010097

**Published:** 2025-01-20

**Authors:** Ying Chen, Junzhe Wang, Zhiwei Miao, Yunpeng Jiang, Xia Wu

**Affiliations:** 1Department of Psychology, Tianjin University of Technology and Education, Tianjin 300222, China; 2Faculty of Psychology, Tianjin Normal University, Tianjin 300387, China

**Keywords:** attentional capture, attentional control settings, target-defining properties, search mode

## Abstract

The attentional control settings (ACSs) can help us efficiently select targets in complex real-world environments. Previous research has shown that category-specific ACS demands more attentional resources than feature-specific ACS. However, comparing natural or alphanumeric categories with color features does not distinguish the effects of processing hierarchy and target-defining properties. The present study employed a spatial cueing paradigm to better understand the effects of target-defining properties and search mode on attentional resources in visual search. The target was defined as a combination of shape feature (shape “X”) and color category (green in different shades), which generated shape-specific ACS (sACS) and color-specific ACS (cACS). The degrees of shape matching (SM), color matching (CM), and spatial validity between the cue and target were manipulated. Search modes were manipulated by changing the homogeneity of distractors in either shape or color dimensions. Results show a main effect of CM across all four experiments, indicating that category can tune on attentional capture consistently. Importantly, the analysis between four experiments found different interactions across experiments, suggesting that the singleton search mode can reduce the effects of ACS and increase the interactions with other factors. In conclusion, this study suggests that the effects of ACS on attentional capture are determined by both target-defining properties and search mode, rather than processing hierarchy. The results indicate that attentional processes are highly dynamic and context-dependent, requiring a flexible allocation of resources to effectively prioritize relevant information.

## 1. Introduction

Attentional capture refers to a process by which task-irrelevant distractors unconsciously attract attention. An increasing number of researchers are focusing on the influence of top-down processing on the attentional capture. [14] ([14]) proposed that attentional control systems can activate target-defining features and efficiently process the matching stimulus. Only stimulus that matches the attentional control settings (ACSs) can capture attention ([14]; [25]). For example, when you are looking for a girl in a red dress in the crowd, the feature of “red” can generate a color-specific ACS and enhance the processing of all stimuli that match “red”. Consequently, the red distractor (e.g., boys wearing red shirt) can attract our attention.

Previous studies have not only explored the impact of top-down processing in a single dimension on attentional capture, but also focused on the more complex information involved in real-world scenarios. Category refers to a set of objects, reflecting a higher hierarchy and more abstractly processing relative to features ([38]). Investigating the role of category information in attentional capture can not only broaden the scope of top-down processing, but also provide theoretical support for visual search in daily situations. In addition to feature-specific ACS, such as presence or absence ([3]; [24]), color ([13]; [27]), and size ([25]), many studies found that category-specific ACS can also play a role on attentional capture ([41]; [19]; [38]; [2]). [40] ([40]), defined the target as a combination of feature and category (e.g., blue letters) to simultaneously examine the integrated role of feature-specific ACS (color) and color-specific ACS (alphanumeric letter). The results show that the feature-specific ACS required fewer attentional resources to operate relative to the category-specific ACS. [39] ([39]) further found that feature-specific ACS weighted larger than category-specific ACS in both attentional enhancement and inhibition.

The weaker strength of category-specific ACS relative to feature-specific ACS may be explained by two reasons. First, previous studies ([40]; [39]) defined color as a feature level and letters as a category level, but the target-defining properties (i.e., the specific perceptual attribute, such as color or shape) may be confused with the hierarchy of feature and category. Previous studies found an advantage of processing for color relative to other properties ([25]; [33]; [1]). Therefore, the findings of [40] ([40]) and [39] ([39]), in which feature-specific ACS required less attentional resources and greater weights than category-specific ACS, may be due to the processing advantage of color attributes itself, rather than differences in the processing of feature and category hierarchies. To address this confound, the present study defined the target as letters with similar colors (green, including dark green, yellow-green, bright green, cyan, and grass green) and a specific shape (“X”), with shape as the feature level and color as the category level. If the effects of shape-specific ACS (sACS) remain greater than that of color-specific ACS (cACS), it further confirms the different weights of feature- and category-specific ACS. Conversely, it will suggest that the results of [40] ([40]) and [39] ([39]) are primarily due to the target-defined properties.

Second, another possible reason for the weaker strength of cACS may be that although the integrate effect of multiple ACSs can operate when the target is defined by a combination of multiple features, these ACSs may have different weights according to dimensional weighting theory ([28]). It is possible that the weight of feature-specific ACS is too great to interfere with the performance of category-specific ACS, and thus only the prominent effect of feature-specific ACS can be observed. If one ACS’s effect can be reduced while the effect of another ACS can be enhanced, it is possible that the two ACSs have interaction effects. In order to better examine the separate effect of each ACS in different search tasks with the same target letter (identifying a target defined by a combination of two attributes, such as a green X), the strength of each ACS can be changed by manipulating different search modes, including feature search mode and singleton search mode ([4]). Feature search mode is based on searches for the specific features of the target, emphasizing the selection of a target with specific attributes, and it is less influenced by distractors. Singleton search mode is based on searches for the pop-out features from the background, emphasizing the exclusion of background distractors, and it is more influenced by distractors. If the target cannot stand out on a certain dimension, that is, other distractors have heterogeneity in that dimension, then only the feature search mode can be used for that dimension, so that only distractors that match the ACS can capture attention, yielding an increased effect of ACS ([12]; [26]; [27]). If the target pops out on a certain dimension, that is, other distractors have homogeneity in that dimension, then the participant is more likely to use the singleton search mode on that dimension, and accordingly, the effect of ACS will decrease ([26]). Therefore, this study manipulated the homogeneity of distractors in shape feature and color category dimensions to change the strength of different ACSs.

To investigate the effects of multiple ACSs and search mode on the search for the same target letter in different search tasks, the present study employed a spatial cueing paradigm and defined the target as a specific shape within a color category (a green “X”). Green was composed of many colors (including dark green, yellow-green, bright green, teal, and grass green), while the shape was unique as an “X”. Three variables were manipulated: the degree of shape-matching degree (SM) between the cue and target as the shape feature level, the degree of color-matching degree (CM) between the cue and target as the color category level, and the validity of the spatial location provided by the cue. By examining the main effects and interaction effects of SM and CM, the strength of the two types of ACS can be explored: if sACS is stronger than cACS, it is consistent with previous research ([40]; [39]; [38]), indicating that the role of ACS is mainly determined by the different processing hierarchies between feature and category, regardless of whether the specific attribute is shape or color. If cACS is stronger than sACS, it indicates that the role of ACS is mainly determined by the shape and color attributes ([25]) rather than the processing levels of feature and category. Additionally, the homogeneity of distractors was manipulated to change the search mode in four experiments. If different search modes have an impact on the strength of ACS, then the patterns and strengths of sACS and cACS in the four experiments may also differ.

## 2. Experiment 1: Color Feature Search and Shape Feature Search

### 2.1. Participants

The sample size calculated by MorePower 6.0 ([7]) indicated that 34 participants would be required to detect an effect size of η^2^p = 0.2 with a power of 0.80 and α level of 0.05 in an 2 × 2 × 2 ANOVA test. Initially, 50 volunteers were recruited from Tianjin Normal University and participated for payment, three subjects were excluded because the accuracies were lower than 60%. The statistical analysis for the remaining 47 participants (14 males) ranging in age from 18 to 29 years (*M* = 21.62, *SD* = 2.16) had been carried out. All had self-reported normal or corrected-to-normal visual acuity and color vision. Each subject volunteered for the study after providing informed consent. The experiment was approved by the Ethics Committee of Tianjin Normal University.

### 2.2. Experimental Design

The design of the experiments was a 2 × 2 × 2 within-subject design, where the independent variables included the degree of shape matching (SM) between cue and target (matching or mismatching), the degree of color matching (CM) between cue and target (matching or mismatching), and the spatial location cue validity of the cue and the target (valid or invalid). The dependent variables were the accuracy of response and the reaction time for correct responses.

### 2.3. Stimuli and Procedure

The experimental program was prepared using E-Prime 2.0 software. The stimuli were presented on a 17-inch CRT monitor with a refresh rate of 100 Hz and resolution of 1024 × 768 pixels. Subjects viewed the screen in a dimly lit room at a distance of approximately 57 cm.

The experiment employed a search task, which involved a target item that is defined by two features (shape and color) and presented among distractors with different combinations of these features. The target in this experiment is a green “X” on a gray background, and the green color includes five different shades of green, including dark green (RGB: 98, 160, 77), yellow green (RGB: 100, 173, 45), bright green (RGB: 50, 180, 50), lime green (RGB: 45, 173, 100), and grass green (RGB: 0, 153, 0), with similar luminance (8.2 cd/m^2^, [21]). The distractors are five different colors, including red (RGB: 145, 0, 0), blue (RGB: 0, 0, 255), purple (RGB: 123, 0, 123), brown (RGB: 88, 63, 0), and gray (RGB: 69, 69, 69), with similar luminance (13 cd/m^2^, [34]) and 15 different shapes (i.e., A, B, D, F, G, H, K, M, N, R, S, V, W, Y, Z were chosen to avoid letters that are similar to X or harder to distinguish).The distractors are randomly combined from these color and shape options, with each combination appearing only once.

As shown in Figure 1A, the experiment started with a 500–1000 ms fixation, followed by a 100 ms cue screen. The cue screen displayed six stimuli of size 1° × 0.8°, located 3.3° away from the central fixation. One of the six positions contained the cue stimulus, while the remaining positions showed black “O” stimuli. The shape and color of cue stimulus were dependent on the degree of shape matching (SM) and color matching (CM) of the current trial and appeared randomly in one of the six positions with equal probability. Specifically, in the S+C+ condition, the cue shape was “X” and the color was one of the five green colors. In the S+C− condition, the cue shape was “X”, and the color was one of the five distractor colors. In the S−C+ condition, the cue shape was “O” and the color was a random color among the five green colors. In the S−C− condition, the cue shape was “O” and the color was a random color among the five distractor colors.

After a 100 ms interval, a 300 ms search screen was displayed. All distractors in the experiment had different colors and shapes from the target. Participants were instructed to quickly and accurately determine the orientation of the frame gap surrounding the green “X” presented on the search screen and press a key in response, and the response window was 1000 ms. The location of the target (green “X”) was determined by the validity of the spatial cue, with the target appearing in the same location as the cue in the valid condition and in a different location from the cue in the invalid condition. The ratio of valid to invalid cues was 1:1. Consequently, the cues were predictive and not classically exogenous. Feedback (500 ms) was provided at the end of each trial. To enhance the role of ACSs in the search task, the experiment added a 1/6 probe test trial, where no target appeared and participants were instructed to respond by pressing the space bar.

Sixteen practice trials followed by a formal experiment consist of 60 trials and 12 probe trials for each condition, yielding a total of 2 × 2 × 2 × (60 + 12) = 576 trials. Participants were allowed a rest after every 5 min, and the experiment lasted approximately 30 min.

### 2.4. Results

The accuracy (*M* ± *SD*) of the probe trials was 93.60 ± 8.67% and the RTs (*M* ± *SD*) was 612 ± 37 ms. The accuracy for different matching and validity conditions (Figure 1B, Appendix A) was subjected to repeated measures of ANOVA with shape matching (SM: S+, S−), color matching (CM: C+, C−), and cue validity (valid, invalid). The results indicate a significant main effect of CM, *F*(1, 46) = 5.50, *p* = 0.023, and η^2^ = 0.11, which showed that the accuracy was lower when the cue matched the color of the target (C+) (91.90 ± 1.70%) than when the color did not match (C−) (93.10 ± 1.70%), suggesting that cACS played a role in the search task. The main effect of validity was significant, *F*(1, 46) = 15.07, *p* < 0.001, and η^2^ = 0.25, with the accuracy of valid cues (93.10 ± 1.70%) being significantly higher than that of invalid cues (91.90 ± 1.70%), indicating a spatial cueing effect.

The interaction between SM and CM was significant, *F*(1, 46) = 4.69, *p* = 0.035, η^2^ = 0.10, and the simple effects analysis revealed a significant difference between C+ and C− when the cue was S−, *F*(1, 46) = 6.81, *p* = 0.012, and η^2^ = 0.13, indicating that sACS and cACS in the search task are not processed independently but have an impact on each other. The interaction between CM and validity was significant, *F*(1, 46) = 5.04, *p* = 0.030, and η^2^ = 0.10, and the simple effects analysis found that the accuracy was significantly smaller when the cue was C+ than C− in invalid condition, *F*(1, 46) = 11.50, *p* = 0.001, and η^2^ = 0.20. A spatial cueing effect was found when the cue was C+, *F*(1, 46) = 16.27, *p* < 0.001, and η^2^ = 0.26, indicating that cACS operates on early attentional allocation.

A repeated-measures ANOVA with SM, CM, and validity for RTs was performed (Figure 1C, Appendix A). A main effect of CM was found, *F*(1, 46) = 120.85, *p* < 0.001, and η^2^ = 0.72, which showed longer RTs in the C+ condition (613 ± 8 ms) relative to the C− condition (594 ± 8 ms), indicating a role for cACS in the search task. The main effect of validity was significant, *F*(1, 46) = 56.07, *p* < 0.001, and η^2^ = 0.55, with the RTs for valid cues (599 ± 8 ms) being significantly shorter than that for invalid cues (607 ± 8 ms), revealing a spatial cueing effect.

In the results of Experiment 1, significant main effects of CM were found on both accuracy and RTs, indicating a larger role for cACS in the search task. Additionally, spatial cueing effects were observed on both accuracy and RTs, indicating a stable early spatial attentional allocation in the spatial cueing paradigm. Although interactions were found between the two ACSs and between cACS and spatial attention on accuracy, no significant interaction was observed on RTs. This suggested that cACS has a stronger power on the RTs index and leads to being less susceptible to other independent factors when both sACS and cACS are in feature search mode.

## 3. Experiment 2: Color Feature Search and Shape Singleton Search

### 3.1. Participants

The sample size calculated by MorePower 6.0 ([7], indicated that 34 participants would be required to detect an effect size of η^2^p = 0.2 with a power of 0.80 and α level of 0.05 in an 2 × 2 × 2 ANOVA test). Initially, 41 volunteers were recruited from Tianjin Normal University and participated for payment, three subjects were excluded because the accuracies were lower than 60%. The statistical analysis for the remaining 38 participants (21 males), ranging in age from 19 to 30 years (*M* = 21.24, *SD* = 1.88), was conducted. All had self-reported normal or corrected-to-normal visual acuity and color vision. Each subject volunteered for the study after providing informed consent. The experiment was approved by the Ethics Committee of Tianjin Normal University.

### 3.2. Experimental Design, Stimuli, and Procedure

Experiment 2 used the same design, stimuli, and procedure as Experiment 1, with the exception that the distractors in the search screen were replaced by colored “O” shapes (Figure 2A). The color of each “O” shape was randomly selected from the following colors: red (RGB: 145, 0, 0), blue (RGB: 0, 0, 255), purple (RGB: 123, 0, 123), brown (RGB: 88, 63, 0), and gray (RGB: 69, 69, 69).

### 3.3. Results

The accuracy (*M* ± *SD*) of the probe trials was 90.55 ± 9.48% and the RTs (*M* ± *SD*) was 686 ± 48 ms. The accuracies for different matching and validity conditions (Figure 2B, Appendix A) were subjected to repeated measures of ANOVA with shape matching (SM: S+, S−), color matching (CM: C+, C−), and cue validity (valid, invalid). The results indicate a significant main effect of CM, *F*(1, 37) = 26.74, *p* < 0.001, and η^2^ = 0.42, which shows that the accuracy was lower when the cue matched the color of the target (C+) (89.00 ± 1.00%) than when the color did not match (C−) (92.60 ± 0.80%), suggesting that cACS played a role in the search task. The main effect of validity was significant, *F*(1, 37) = 8.16, *p* = 0.007, and η^2^ = 0.18, with the accuracy of valid cues (91.30 ± 0.80%) being significantly higher than that of invalid cues (90.30 ± 0.90%), indicating a spatial cueing effect. The interaction between CM and validity was marginally significant, *F*(1, 37) = 3.29, *p* = 0.078, and η^2^ = 0.08, and the simple effects analysis revealed a significant difference between C+ and C−, regardless of the validity of the cue [valid: *F*(1, 37) = 12.92, *p* = 0.001,η^2^ = 0.26; invalid: *F*(1, 37) = 23.87, *p* < 0.001, and η^2^ = 0.39]. A spatial cueing effect was found with C+ cues, *F*(1, 37) = 6.88, *p* = 0.013, and η^2^ = 0.16, but not with C− cues (*p* = 0.717), indicating that cACS played a role in early attentional orientation.

A repeated-measures ANOVA with SM, CM, and validity for RTs was performed (Figure 2C, Appendix A). A main effect of CM was found, *F*(1, 37) = 59.48, *p* < 0.001, and η^2^ = 0.62, which showed longer RTs in C+ condition (677 ± 9 ms) relative to C− condition (659 ± 8 ms), indicating a role for cACS in the search task. The main effect of validity was significant, *F*(1, 37) = 22.93, *p* < 0.001, and η^2^ = 0.38, with the RTs for valid cues (665 ± 9 ms) being significantly shorter than that for invalid cues (670 ± 9 ms), revealing a spatial cueing effect. The interaction between SM and CM was significant, *F*(1, 37) = 16.02, *p* < 0.001, and η^2^ = 0.30, and the simple effects analysis revealed longer RT of S− than that of S+ when the cue was C+, *F*(1, 37) = 12.35, *p* < 0.001, and η^2^ = 0.25, whereas the trend was the opposite in the C− condition: the RT of S+ was longer than that of S−, *F*(1, 37) = 4.51, *p* = 0.040, and η^2^ = 0.11. The RT of the C+ was significantly greater than that of the C−, regardless of the shape of the cue [S+: *F*(1, 37) = 21.87, *p* < 0.001, η^2^ = 0.37; S−: *F*(1, 37) = 81.77, *p* < 0.001, and η^2^ = 0.69], indicating that sACS and cACS were not processed independently, but interfering with each other.

In the results of Experiment 2, similar to Experiment 1, significant main effects of CM were found on both accuracy and RTs, indicating the critical role for cACS in the search task. In addition, spatial cueing effects were found on both accuracy and RTs. However, unlike Experiment 1, an interaction between SM and CM was found in RTs when sACS was in the singleton search mode and cACS was in the feature search mode. This indicated that the enhanced cACS could interfere with sACS, suggesting that multiple ACSs can exist simultaneously and interact with each other in the search task.

## 4. Experiment 3: Color-Singleton Search and Shape-Feature Search

### 4.1. Participants

The sample size calculated by MorePower 6.0 ([7]) indicated that 34 participants would be required to detect an effect size of η^2^p = 0.2 with a power of 0.80 and α level of 0.05 in an 2 × 2 × 2 ANOVA test. Initially, 45 volunteers were recruited from Tianjin Normal University and participated for payment, 5 subjects were excluded because the accuracies were lower than 60%. The statistical analysis for the remaining 40 participants (7 males) that ranged in age from 18 to 25 years (*M* = 20.28, *SD* = 1.20) had been conducted. All had self-reported normal or corrected-to-normal visual acuity and color vision. Each subject volunteered for the study after providing informed consent. The experiment was approved by the Ethics Committee of Tianjin Normal University.

### 4.2. Experimental Design, Stimuli, and Procedure

Experiment 3 used the same design, stimuli, and procedure as Experiment 1, with the exception that the distractors in the search screen were replaced by a single color with different shapes (Figure 3A). The color and shapes were randomly selected from the following features: red (RGB: 145, 0, 0), blue (RGB: 0, 0, 255), purple (RGB: 123, 0, 123), brown (RGB: 88, 63, 0), and gray (RGB: 69, 69, 69); A, B, D, F, G, H, K, M, N, R, S, V, W, Y, and Z.

### 4.3. Results

The accuracy (*M* ± *SD*) of the probe trials was 95.24 ± 4.51% and of the RTs (*M* ± *SD*) was 612 ± 64 ms. The accuracies for different matching and validity conditions (Figure 3B, Appendix A) were subjected to repeated measures of ANOVA with shape matching (SM: S+, S−), color matching (CM: C+, C−), and cue validity (valid, invalid). The results indicate a significant main effect of CM, *F*(1, 39) = 4.28, *p* = 0.045, and η^2^ = 0.10, which shows that the accuracy was lower when the cue matched the color of the target (C+) (92.60 ± 0.80%) than when the color did not match (C−) (93.70 ± 0.80%), suggesting that cACS played a role in the search task. The main effect of validity was significant, *F*(1, 39) = 11.46, *p* = 0.002, and η^2^ = 0.23, with the accuracy of valid cues (93.80 ± 0.70%) being significantly higher than that of invalid cues (92.40 ± 0.80%), indicating a spatial cueing effect. The interaction between CM and validity was significant, *F* (1, 39) = 7.78, *p* = 0.008, and η^2^ = 0.17, and the simple effects analysis revealed a significant difference between C+ and C− when the cue was invalid, *F*(1, 39) = 10.54, *p* = 0.002, and η^2^ = 0.21. A spatial cueing effect (higher accuracy in valid condition relative to invalid condition) occurred when the cue was C+, *F*(1, 39) = 14.89, *p* < 0.0 01, and η^2^ = 0.28, but absence when the cue was C− (*p* = 0.586), indicating that cACS operates on early attentional orientation.

A repeated-measures ANOVA with SM, CM, and validity for RTs was performed (Figure 3C, Appendix A). A main effect of CM was found, *F*(1, 39) = 103.85, *p* < 0.001, and η^2^ = 0.73, which showed longer RTs in C+ condition (599 ± 10 ms) relative to C− condition (582 ± 9 ms), indicating a role for cACS in the search task. The main effect of validity was significant, *F*(1, 39) = 42.24, *p* < 0.001, and η^2^ = 0.52, with the RTs for valid cues (587 ± 9 ms) being significantly shorter than that for invalid cues (594 ± 9 ms), revealing a spatial cueing effect. The interaction between CM and validity was significant, *F*(1, 39) = 10.46, *p* = 0.002, and η^2^ = 0.21, and the simple effects analysis revealed a significant difference between valid and invalid, regardless of the color of the cue [C+: *F*(1, 39) = 39.69, *p* < 0.001, η^2^ = 0.50; C −: *F*(1, 39) = 15.31, *p* < 0.001, and η^2^ = 0.28]. The RT of the C+ was significantly greater than that of the C−, regardless of the validity of the cue (valid: *F*(1, 39) = 52.30, *p* < 0.001, η^2^ = 0.57; invalid: *F*(1, 39) = 118.75, *p* < 0.001, and η^2^ = 0.75), indicating that cACS operates on early attentional orientation.

In the results of Experiment 3, similar to Experiments 1 and 2, significant main effects of CM were found on both accuracy and RTs, indicating the stable effects of cACSs in the search task. Spatial cueing effects were also observed on both accuracy and RTs. Unlike Experiments 1 and 2, when sACS was in the feature search mode and cACS was in the singleton search mode, an interaction between CM and validity was found on both accuracy and RTs, suggesting that the reduced cACS was more susceptible to early spatial attention.

## 5. Experiment 4: Color-Singleton Search and Shape-Singleton Search

### 5.1. Participants

The sample size calculated by MorePower 6.0 ([7]) indicated that 34 participants would be required to detect an effect size of η^2^p = 0.2 with a power of 0.80 and α level of 0.05 in an 2 × 2 × 2 ANOVA test. Initially, 50 volunteers were recruited from Tianjin Normal University and participated for payment, 4 subjects were excluded because the accuracies were lower than 60%. The statistical analysis for the remaining 46 participants (9 males) ranging in age from 18 to 24 years (*M* = 20.67 *SD* = 1.03) had been conducted. All had self-reported normal or corrected-to-normal visual acuity and color vision. Each subject volunteered for the study after providing informed consent. The experiment was approved by the Ethics Committee of Tianjin Normal University.

### 5.2. Experimental Design, Stimuli, and Procedure

Experiment 4 used the same design, stimuli, and procedure as Experiment 1, with the exception that the distractors in the search screen became a single color “O” (Figure 4A). The color of each “O” shape was randomly selected from the following colors: red (RGB: 145, 0, 0), blue (RGB: 0, 0, 255), purple (RGB: 123, 0, 123), brown (RGB: 88, 63, 0), and gray (RGB: 69, 69, 69).

### 5.3. Results

The accuracy (*M* ± *SD*) of the probe trials was 92.74 ± 8.44% and of the RTs (*M* ± *SD*) was 623 ± 71 ms. The accuracy for different matching and validity conditions (Figure 4B, Appendix A) were subjected to repeated measures ANOVA with shape matching (SM: S+, S−), color matching (CM: C+, C−) and cue validity (valid, invalid). The results indicate a significant main effect of SM, *F*(1, 45) = 1.79, *p* = 0.188, and η^2^ =0.04. The main effect of CM was significant, *F*(1, 45) = 13.22, *p* = 0.001, and η^2^ = 0.23, which showed that the accuracy was lower when the cue matched the color of the target (C+) (93.10 ± 1.10%) than when the color did not match (C−) (94.50 ± 1.00%), suggesting that cACS played a role in the search task. The main effect of validity was significant, *F*(1, 45) = 10.11, *p* = 0.003, and η^2^ = 0.18, with the accuracy of valid cues (94.20 ± 1.00%) being significantly higher than that of invalid cues (93.30 ± 1.00%), indicating a spatial cueing effect. The interaction between SM and CM was not significant, *F*(1, 45) = 1.56, *p* = 0.22, and η^2^ = 0.03. The interaction between SM and validity was significant, *F*(1, 45) = 7.44, *p* = 0.009, and η^2^ = 0.14, and the simple effects analysis revealed a significant difference between S+ and S− when the cue was valid, *F*(1, 45) = 6.98, *p* = 0.011, and η^2^ = 0.13, with a significant difference between valid and invalid when the cue was S−, *F*(1, 45)= 22.63, *p* < 0.001, and η^2^ = 0.33, indicating that sACS played a role in early attentional orientation. The interaction between CM and validity was significant, *F*(1, 45) = 6.44, *p* = 0.015, and η^2^ = 0.13, and the simple effects analysis found that the accuracy was significantly higher when the cue was valid than invalid in the C+ condition, *F*(1, 45) = 14.01, *p* = 0.001, and η^2^ = 0.24, indicating that the role cACS played in early attentional orienting differs from the way of sACS. The interaction between SM, CM, and validity was significant, *F*(1, 45) = 10.47, *p* = 0.002, and η^2^ = 0.19, and simple effects analysis reveals that a spatial cuing effect only occurred in the condition of S−C+, *F*(1, 45) = 33.11, *p* < 0.001, and η^2^ = 0.42.

A repeated-measures ANOVA with SM, CM, and validity for RTs was performed (Figure 4C, Appendix A). A main effect of SM was found, *F*(1, 45) = 3.84, *p* = 0.056, and η^2^ = 0.08, which showed longer RTs in the S− condition (612 ± 11 ms) relative to S+ condition (610 ± 11 ms), indicating that sACS played a role in the search task. The CM main effect was significant, *F*(1, 45) = 18.75, *p* < 0.001, and η^2^ = 0.29, which showed longer RTs in the C+ condition (615 ± 11 ms) relative to C− condition (607 ± 12 ms), indicating a role for cACS in the search task. The main effect of validity was significant, *F*(1, 45) = 42.28, *p* < 0.001, and η^2^ = 0.48, with the RTs for valid cues (608 ± 11 ms) being significantly shorter than that for invalid cues (615 ± 11 ms), revealing a spatial cueing effect. The interaction between SM and CM was significant, *F*(1, 45) = 16.56, *p* < 0.001, and η^2^ = 0.27, and the simple effects analysis revealed a longer RTs of S− than that of S+ when the cue was C+, *F*(1, 45) = 17.51, *p* < 0.001, and η^2^ = 0.28, whereas the trend was the opposite in the C− condition: the RT of S+ was longer than that of S−, *F*(1, 45) = 10.18, *p* = 0.003, and η^2^ = 0.18. A significant difference was found between C+ and C− when the cue was S−, *F*(1, 45)= 48.44, *p* < 0.001, and η^2^ = 0.52. The results suggest that sACS and cACS were not processed independently, but compete with each other in the search task. The interaction between SM, CM, and validity was significant, *F*(1, *45*) = 4.71, *p* = 0.035, and η^2^ = 0.09, and simple effects analysis reveals that cues in all matching conditions triggered a spatial cueing effect except when the cue was S−C− (*p* = 0.351). S+ C+: *F*(1, 45) = 5.44, *p* = 0.024, η^2^ = 0.11; S+C−: *F*(1, 45) = 11.46, *p* = 0.001, η^2^ = 0.20; S−C+: *F*(1, 45) = 21.67, *p* < 0.001, and η^2^ = 0.33, indicating a top-down effect of processing on attentional capture.

In the results of Experiment 4, similar to Experiments 1, 2, and 3, significant main effects of CM were found on both accuracy and RTs, indicating the stable effects of cACS in the search task. Spatial cueing effects were also found on both accuracy and RTs, indicating the stable early spatial attentional orienting. Unlike Experiments 1, 2, and 3, when both sACS and cACS were in singleton search mode, the interaction between SM and CM, the interaction between CM and validity, and the interaction between SM, CM, and validity were all found on both accuracy and RTs. These findings suggest that the search mode can affect the strength of the ACS. When the effects of both ACSs are reduced, they are more likely to interact with each other, spatial attention is more likely to affect cACS, and the interaction between sACS, cACS, and spatial attention is also more likely to be observed.

## 6. Overall Comparison

To compare the *p* value of main effects and their interactions across four experiments, two heatmaps were illustrated for accuracy (Figure 5A) and RTs (Figure 5B). From the graph, it can be observed that the effect of cACS (the main effect of CM) was relatively stable, as well as the spatial cueing effects (the main effect of validity). Additionally, when both color and shape are in the singleton search mode (Experiment 4), the number of interactions increases. The comparison of *p* value heatmaps indicated that the singleton search mode can decrease the strength of multiple ACSs and lead them to being more susceptible to various interactions.

Furthermore, to investigate the roles of cACS and sACS on the spatial attentional orientation across different search modes, a mixed ANOVA, with SM(S+, S−), CM(C+, C−), and validity (valid, invalid) as the within-subject factor and SS (shape search mode: feature, singleton), and CS (color search mode: feature, singleton) as the between subject factor, was performed. The results of accuracy show a significant main effect of CM, *F*(1, 167) = 46.60, *p* < 0.001, and η^2^ = 0.218, showing decreased accuracy of target identification when the cue was matching the targets’ color, a main effect of validity, *F*(1, 167) = 44.23, *p* < 0.001, and η^2^ = 0.209, showing a stable spatial cueing effect. The interaction between CM and SS [*F*(1, 167) = 6.07, *p* = 0.015, and η^2^ = 0.035], CM and CS [*F*(1, 167) = 4.88, *p* = 0.029, and η^2^ = 0.028], show that the cACS worked no matter the search modes (*ps* < 0.002). The interaction between SM and CM [*F*(1, 167) = 4.91, *p* = 0.028, and η^2^ = 0.029], CM and validity [*F*(1, 167) = 21.69, *p* < 0.001, and η^2^ = 0.115], and SM, CM, and validity [*F*(1, 167) = 5.37, *p* = 0.022, and η^2^ = 0.031], showed similar results in a separate experiment. Importantly, the significant interaction among SM, CM, validity, and SS [*F*(1, 167) = 5.49, *p* = 0.020, and η^2^ = 0.032] indicated that the effects of sACS and cACS on spatial attentional orientation are influenced by the shape search mode.

The results of mixed ANOVA for RTs show a significant main effect of CM, *F*(1, 167) = 265.93, *p* < 0.001, and η^2^ = 0.614, showing a slower reaction of target identification when the cue was matching the targets’ color, a main effect of validity, *F*(1, 167) = 157.50, *p* < 0.001, and η^2^ = 0.485, showing a stable spatial cueing effect, a main effect of SS, *F*(1, 167) = 20.68, *p* < 0.001, and η^2^ =0.110, showing faster reaction in the shape feature search mode relative to shape singleton search mode, and a main effect of CS, *F*(1, 167) = 13.39, *p* < 0.001, and η^2^ = 0.074, showing a reverse trend to SS. The interactions between CM and SS [*F*(1, 167) = 6.74, *p* = 0.010, and η^2^ = 0.039], and CM and CS [*F*(1, 167) = 11.05, *p* = 0.001, and η^2^ = 0.062], show that the cACS worked no matter what search modes (*ps* < 0.001). The interactions between SM and CM [*F*(1, 167) = 24.86, *p* < 0.001, and η^2^ = 0.130], CM and validity [*F*(1, 167) = 11.91, *p* = 0.001, and η^2^ = 0.067], and SM, CM, and validity [*F*(1, 167) = 4.95, *p* = 0.027, and η^2^ = 0.029] were significantly similar as shown above in a separate experiment. Importantly, the significant interaction among SM, CM, and SS [*F*(1, 167) = 12.99, *p* < 0.001, and η^2^ = 0.072] indicated that the effects of sACS and cACS can be integrated and influenced by the shape search mode.

## 7. Discussion

To investigate the effects of target-defining properties and search modes on the strength of multiple attentional control settings (ACSs) in attentional capture, a spatial cuing paradigm in which targets were defined as complex objects with a combination of shape features and color categories was employed. The effects of shape-specific ACS (sACS) and color-specific ACS (cACS) on attentional capture were examined by manipulating the match level between the pre-target cues and the shape (SM), color (CM), and spatial position (validity) of the target. In addition, four experiments were conducted to control the homogeneity of the distractors in shape and color, respectively, to examine the effects of search mode on the strength of the two ACS. The main findings are as follows: Firstly, target-defining attributes determined the strength of multiple ACSs. Regardless of the search mode used, cues interfered with target selection when they matched with the cACS, indicating that color was a stronger attribute in the visual search and played an important role in attentional capture. The findings suggest that the influence of multiple ACSs on attentional capture is determined by perceptual features, such as color, size, or shape, rather than the hierarchical information of the ACSs. Secondly, early spatial attention played an important role in the spatial cueing paradigm, which is indicated by the result that target selection and response were faster when cues matched the target’s locations, regardless of the search mode used. Lastly, search mode could affect the strength of multiple ACSs. When a singleton search mode was used, the strength of the cACS became weaker and more susceptible to other factors, such as an interaction between sACS and cACS, an interaction between cACS and early spatial attention, and a three-way interaction among sACS, cACS, and spatial attention. Overall, the present study highlights the importance of target-defining properties and search modes in the role of multiple ACSs in attentional capture.

A target-defining attribute, rather than processing hierarchy, was the main factor in influencing the effects of ACS. Target-defining attributes refer to the specific perceptual feature pathways that define a target (i.e., shape and color, etc.), and processing hierarchy refers to the hierarchy of information processing that defines the target, with features being low-level dimensions and categories being high-level dimensions. Unlike previous studies, the present study used color as a category hierarchy, defining the target as “green”, which contains different specific types of green (dark green, yellow green, bright green, lime green, and grass green), and found that cACS played a strong role in attentional capture (evident of a main effect of cACS in all four experiments) ([21]; [10]). The results also confirm that color attributes have a processing priority, enabling people to process color better than size ([25]), shape ([11]), and symbolic category ([31]) in attentional capture, and can also increase search efficiency in realistic scenarios ([31]). More importantly, the priority of color processing could be extended so that it works not only as feature hierarchy, but also as category hierarchy in the attentional capture. The results suggest that the difference in attentional resource requirements and processing weights between cACS and sACS of [40] ([40]) and [39] ([39]) was caused by color and shape properties themselves, rather than the processing hierarchy. Notably, the results of the present study differ from previous research ([43]; [38]; [40]) and suggest that the difference in the strength of color and shape overwhelmed the discrepancy between feature and category dimensions in attentional capture. Future research could investigate the impact of both feature and category dimensions on the same attribute (e.g., color) on attentional capture.

By varying the search pattern employed in different ACSs through four experiments, the present study found that the search pattern can change the strength of ACS but does not affect the early spatial attention. The singleton search mode reduced the strength of ACS, making it more susceptible to other factors, while the feature search mode increased the strength of ACS and made it less susceptible to other factors. The results are in line with the hypothesis of search mode theory ([4]), which suggested that the singleton search pattern relies more on bottom-up processing, while the feature search mode relies more on top-down processing. As a result, the influence of cACS on early spatial attention may be weaker in the singleton search mode compared to the feature search mode. This supports the idea that the strength of top-down control in a visual search task depends on the search mode employed by the observer. However, the role of cACS was still mainly manifested in Experiment 2 and Experiment 3 when a singleton search mode was applied to shape and color alone due to the overweighting of the color attribute, indicating that target-defining attributes are the fundamental determinants of the action of ACSs when multiple ACSs work together, regardless of the changing strength of ACSs through different search patterns. In addition, we found that the stable spatial cueing effect (i.e., the role of early spatial attentional orientations) was not significantly different between four experiments, indicating that spatial attention was not influenced by the search paradigm. This supports the idea that spatial and feature attention play distinct roles in the perceptual cortex ([15]; [32]). Except the separate roles of spatial and feature attention, the interaction between them in the posterior parietal cortex ([17]) and their integrated role originating from the top-down processing ([32]) have become a topic of increasing interest among researchers.

The results of Experiment 2 suggest that the interaction between sACS and cACS occurs in the two-stage selection scenario in a visual search ([25]; [41]). In the early stage, multiple ACSs operate independently according to their respective pathways. In the late target selection stage, sACS and cACS interact and integrate with each other according to the weights of the relevant attributes (shape and color) in the activation map. When the cue matched with cACS (C+), the role of sACS decreased and a reversed capture effect occurred. However, when the cue did not match with cACS (C−), sACS functioned normally, while the role of cACS was not affected by sACS. These findings are consistent with previous studies ([18]; [25]; [11]; [31]; [31]; [40], [41]; [39]) and support the guided search model ([36], [37]), which postulated that multiple perceptual pathways form an activation map that is used to filter subsequent attentional selection and control, and top-down and bottom-up processing will collaborate to form the activation map. The results also highlight the importance of considering the interaction between different ACSs in search tasks.

The interaction between cACS and early spatial attention was found in Experiment 3, where the strength of cACS decreased when participants employed singleton search mode in the color dimension. Specifically, cACS influenced early spatial attention when distractors were homogeneous of color, and spatial attention could reversely enhance cACS. These findings are in line with previous research that used multivariate classifiers to explore the effect of ACS in real scenes ([22]; [5]) and suggest that category-based information, whether color or object, can holistically direct attention from top-down and occur prior to early spatial attentional orienting ([20]). The results also support the idea that categories can not only integrate received perceptual features to form abstract and conceptual information ([42]), but also work independently to influence attentional capture during the preattentive phase ([35]; [16]).

In the present study, the results of the main effect of CM show that matching cues result in slower RTs of target identification relative to nonmatching ones. However, Moore and colleagues found that placing an irrelevant color into memory facilitated subsequent responses to the same stimuli ([29]; [30]). The discrepancy between the findings of the current study and those of Moore and colleagues can be attributed to differences in experimental design and the cognitive mechanisms involved. The results of Moore and colleagues suggest that memory maintenance can create a cognitive advantage for processing familiar stimuli. In contrast, this study focuses on the immediate effects of color matching on attentional capture during a visual search task. When the cue matches the target color, the cACS is strongly activated, leading to a more intense capture of attention. This intense capture may interfere with other cognitive processes, resulting in slower RTs as the system takes longer to integrate and filter information ([14]; [25]). The difference in outcomes highlights the complexity of attentional processes, where memory-based facilitation and attentional capture can have opposing effects depending on the task demands and cognitive resources allocated. This study’s findings emphasize that in complex visual search tasks, strong attentional capture does not always translate to faster response times, potentially due to increased cognitive load and the need for additional processing to manage the captured attention.

While this study examined the effects of color matching on attentional capture, it did not delve into the precision of attentional tuning to color variation ([23]). Future research could explore how varying levels of color similarity impact the precision and efficiency of attentional selection, providing insights into the mechanisms of attentional tuning in complex visual tasks. Additionally, further studies may be necessary to extend the present results to neuroimaging evidence, such as the dorsal attentional network (superior parietal gyrus), anterior parietal salience network (anterior insula, anterior cingulate gyrus, and middle frontal gyrus) ([6]), and top-down attentional control networks (ventral control system, inferior frontal gyrus, joint parietotemporal area, etc.; [9]; [8]). Moreover, the study only manipulated the color and shape dimensions, while other dimensions, such as texture, size, and orientation, may also have a significant impact on attentional capture, and their interaction with multiple ACSs should be studied in future research. Finally, future studies may explore the influence of ACS on attentional capture in individuals with neuropsychological disorders, such as attention deficit hyperactivity disorder (ADHD) or anxiety disorder.

In summary, the present study investigated the effect of multiple ACSs on attentional capture in visual search. The results indicate that the target definition attribute is the main factor that determined whether multiple ACSs would be activated, with the color attribute having a stronger effect than the shape attribute. The search modes can affect the strength of ACSs in a visual search, and both ACSs can interact with each other in a two-stage selection scenario. In addition, category-based information, such as color, can influence attentional capture holistically. The results challenge our understanding of how attention operates in real-world environments, as well as the role of target-defining properties and search mode in attentional capture.

## Figures and Tables

**Figure 1 behavsci-15-00097-f001:**
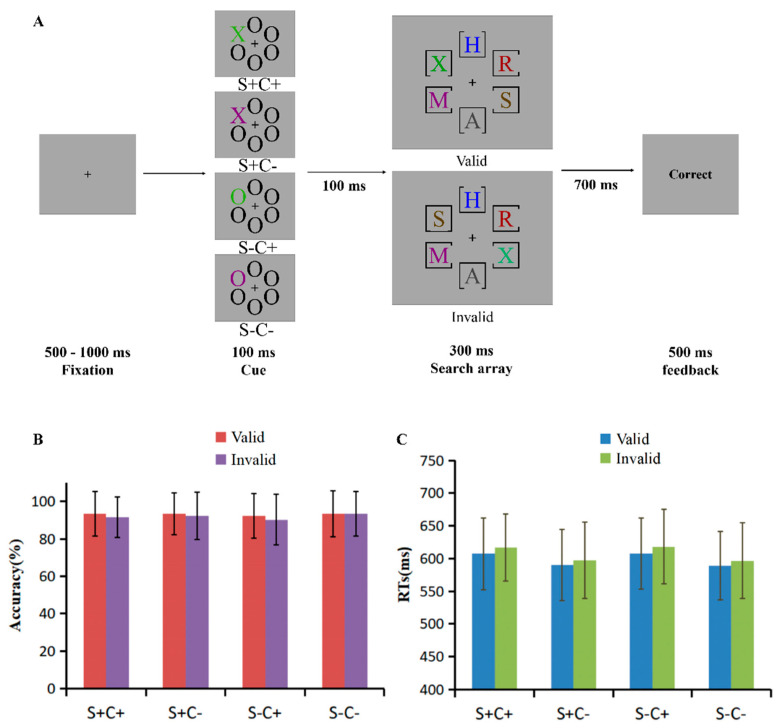
(**A**) Schematic representation of the procedure. After the fixation, a cue screen appears with four levels of match to the defined properties of the target: S+C+, shape and color match; S+C−, shape match, color mismatch; S−C+, shape mismatch, color match; and S−C−, shape and color mismatch. After a 100 ms interval, a search screen appeared and participants were asked to report the orientation of the frame gap surrounding the “green X” in the search screen quickly and accurately, with a response window of 1000 ms. The location of the target could be consistent with the cue (valid cue) or inconsistent with the cue (invalid cue). Distractors other than the target were different colored and shaped stimuli. The location of the cue and the orientation of the frame gap were randomized. (**B**) Accuracy (%) (*M* ± *SD*) and (**C**) reaction times (ms) (*M* ± *SD*) of matching level (S+C+, S+C−, S−C+, and S−C−) and cue validity (valid, invalid) are found in Experiment 1.

**Figure 2 behavsci-15-00097-f002:**
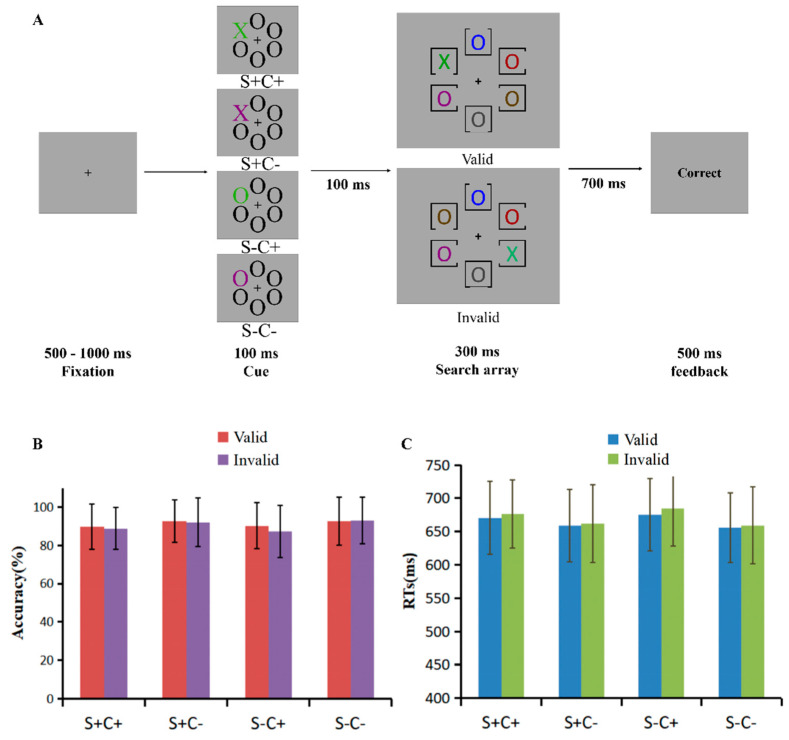
(**A**) The schematic representation of the procedure of Experiment 2. (**B**) Accuracy (%) (*M* ± *SD*) and (**C**) reaction times (ms) (*M* ± *SD*) of matching level (S+C+, S+C−, S−C+, and S−C−) and cue validity (valid, invalid) in Experiment 2.

**Figure 3 behavsci-15-00097-f003:**
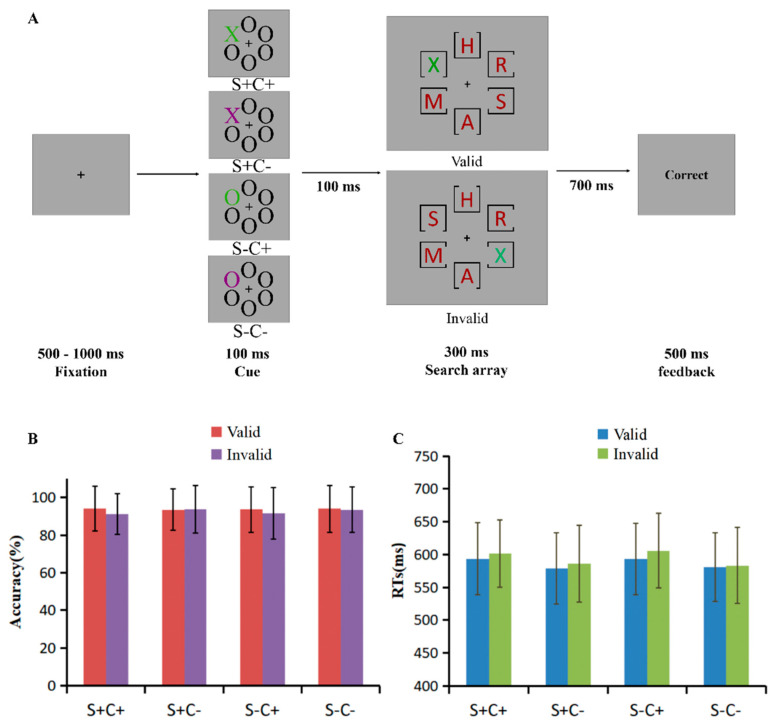
(**A**) The schematic representation of the procedure of Experiment 3. (**B**) Accuracy (%) (*M* ± *SD*) and (**C**) reaction times (ms) (*M* ± *SD*) of matching level (S+C+, S+C−, S−C+, and S−C−) and cue validity (valid, invalid) in Experiment 3.

**Figure 4 behavsci-15-00097-f004:**
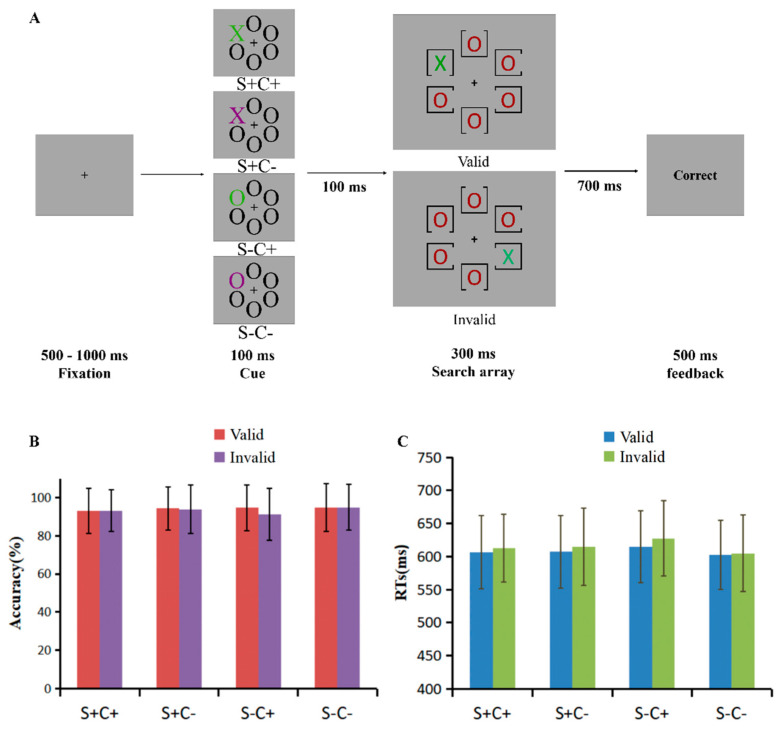
(**A**) The schematic representation of the procedure of Experiment 4. (**B**) Accuracy (%) (*M* ± *SD*) and (**C**) reaction times (ms) (*M* ± *SD*) of matching level (S+C+, S+C−, S−C+, and S−C−) and cue validity (valid, invalid) in Experiment 4.

**Figure 5 behavsci-15-00097-f005:**
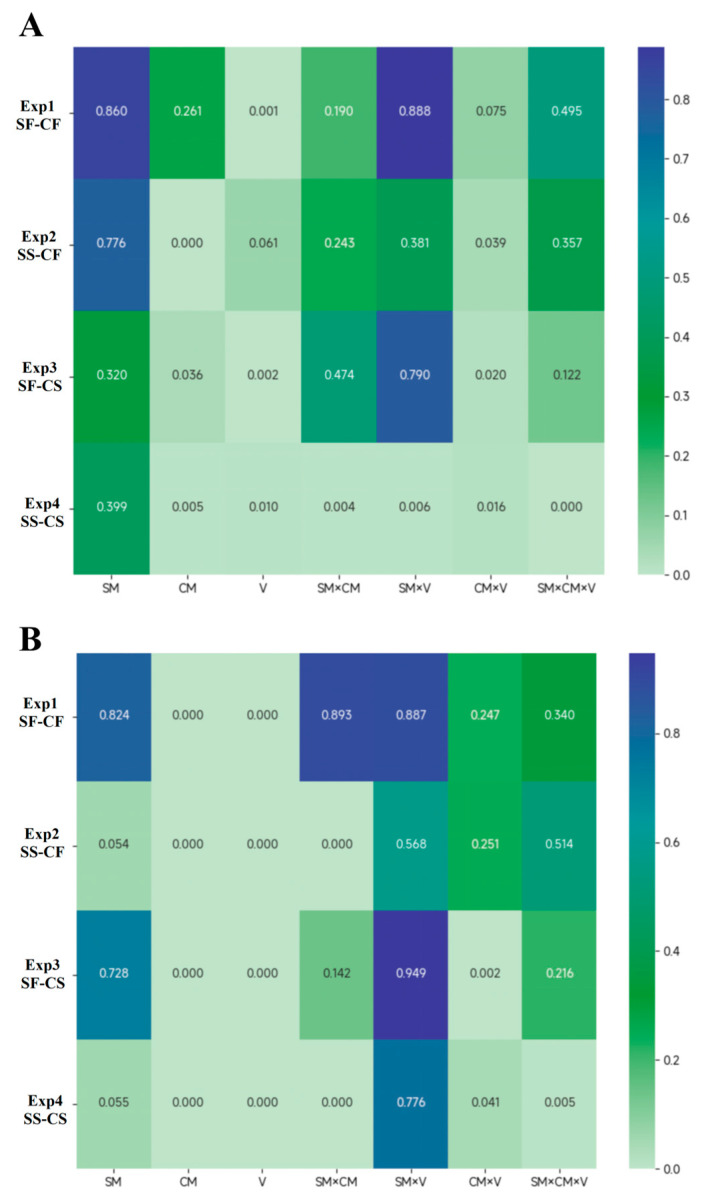
Heatmaps of *p*-values for the main effects of SM, CM, and validity, as well as their three-way interaction, across four experiments. (**A**) Accuracy (%) and (**B**) reaction times (ms). “V” represents validity and the symbol “×” represents interaction. SF: shape feature search mode; CF: color feature search mode; SS: shape singleton search mode; and CS: color singleton search mode.

## Data Availability

Data are available at https://osf.io/ar6j7/ (open data-link).

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
