# Peer review of "The Target-Defining Attributes Can Determine the Effects of Attentional Control Settings in Singleton Search Mode"

_behavsci, 2025, doi:10.3390/bs15010097_

Round 1
Reviewer 1 Report
Comments and Suggestions for Authors
The authors manipulated the type of search and the type of target in four experiments. They find that correspondence between the colors of cues preceding the search display decreases performance. There were several interactions involving cue validity, suggesting that effects of correspondence are weaker in singleton detection mode.
The authors present an interesting series of experiments. However, there are some errors in the terminology that need to be correct (e.g., “conjunction search”). Also, the results are difficult to follow, and I suggest running a between-experiment ANOVA.
The search task is described as conjunction search, but this is not true. In all four experiments, the color and the shape of the target were unique. That is, the target was a green X, but there were no green H or blue X, for example. Thus, it wasn’t necessary to look for a conjunction, but for one of two fixed features. For instance, in Treisman’s studies, there were two feature dimensions (color and shape) and the target was a unique combination of features on the two dimensions (Treisman & Gelade, 1980). For instance, the colors were brown and green, and the shapes were T and X. If the target was a green T, then there would be brown T and a green X. This logic is different from the current experiments. Therefore, the term “conjunction search” is wrong. Rather, there were different combinations of feature and singleton search (according to the terminology in Bacon & Egeth, 1994).
The experiments follow a factorial design where feature or singleton search is performed along one or two dimensions. When color is indicated by C and shape by S, whereas feature search is indicated by F and singleton search by S, then the design between experiments is clear:
E1: Color: Feature search, Shape: Feature search: C-F, S-F
E2: Color: Feature search, Shape: Singleton search, C-F, S-S
E3: Color: singleton search, Shape: Feature search, C-S, S-F
E4: Color: singleton search, Shape: Singleton search, C-S, S-S
This is a very interesting design, and the authors may want to combine the four experiments into a single experiment with four groups. The analysis should be a mixed-factors ANOVA where the between-subject factors 2 (type of shape search: feature, singleton) x 2 (type of color search: feature, singleton) should be included. The within-subject factors should be as before. What are the predictions for this design? In a way, the authors already tested it, but the hypotheses are difficult to extract because they are spread out across four experiments. The new analysis would allow for a much concise manuscript.
Figure 2 and following: the bar graphs for the errors are tedious to read. Maybe it would be better to have two separate line plots side by side. One for RTs and the other for errors.
The authors find that matching colors result in slower RTs than nonmatching colors (main effect of CM, e.g., line 207). Moore and colleagues, however, found that placing an irrelevant color into memory facilitated subsequent responses to the same stimuli (Moore & Weissman, 2011; Moore & Weissman, 2014). The authors should discuss why they find the opposite. That is, worse performance with matching colors.
I did not quite understand why there were different shades of the target color (green). What was the cue color? The same shade of green? It has been shown that attentional tuning to color is narrower in feature search than in singleton search (Kerzel, 2019), however, it is unclear whether effects of color variation were evaluated at all in the current study. Please clarify.
line 109: It is not clear to me what is meant by “processing levels of feature and category”. Please explain the terminology
line 116: A justification for the sample size is missing.
Line 174: If the proportion of valid to invalid cues was 1:1, then the cue predicted the target location. If cue and target location were selected randomly, a ratio of valid to invalid cues of 1:6 would be expected. The authors need to state that the cues were predictive and therefore, not classically “exogenous”.
Line 251: The authors say the effect was significant, but the p-value was .078. The threshold of significance is .05. Therefore, it would be better to write that the effect approached significance.
Line 254-255: The authors test for differences between valid and invalid for C+ and C-, but there was no interaction between validity and color match. Therefore, there shouldn’t be any post-hoc comparisons.
Lines 307-310: For the interaction of CM and validity, the authors should test whether the validity effect was significant both for C- and C+. The current t-tests are a mixture of several questions and should be removed.
Line 317-324: The authors find an interaction of CM and validity. The posthoc tests in the text do not match the graphs. The authors write that regardless of the color of the cue, there was a difference between valid and invalid. However, this is not true when looking at Figure 5 where only the C+ conditions result in cueing effects. Please clarify. Similarly, the effect of CM appears reliable for valid trials, but not invalid trials (lines 322-324). Please clarify.
Line 362: With a p-value of .22, the interaction between SM and CM was not significant. The authors need to revise the sentence saying that it was significant.
Lines 392-393: “in the conjunction search task.” This is wrong because the search task was not a conjunction search task. The target was both a shape and a color singleton and the specific combination of shape was not necessary to find it.
Bacon, W. F., & Egeth, H. E. (1994). Overriding stimulus-driven attentional capture. Perception & Psychophysics, 55(5), 485-496. https://doi.org/10.3758/BF03205306
Kerzel, D. (2019). The precision of attentional selection is far worse than the precision of the underlying memory representation. Cognition, 186, 20-31. https://doi.org/10.1016/j.cognition.2019.02.001
Moore, K. S., & Weissman, D. H. (2011). Set-specific capture can be reduced by pre-emptively occupying a limited-capacity focus of attention. Visual Cognition, 19(4), 417-444. https://doi.org/10.1080/13506285.2011.558862
Moore, K. S., & Weissman, D. H. (2014). A bottleneck model of set-specific capture. PloS One, 9(2), e88313. https://doi.org/10.1371/journal.pone.0088313
Treisman, A. M., & Gelade, G. (1980). A feature-integration theory of attention. Cognitive Psychology, 12(1), 97-136. https://doi.org/10.1016/0010-0285(80)90005-5
Author Response
Response to Reviewer 1 Comments
|
||
1. Summary |
|
|
Thank you very much for taking the time to review our manuscript. We sincerely appreciate your constructive comments and suggestions. Below, you will find our detailed responses along with the corresponding revisions, highlighted in green in the resubmitted manuscript. We have made every effort to address your concerns thoughtfully, and we value your contribution to improving the quality of this work.
|
||
2. Point-by-point response to Comments and Suggestions for Authors |
||
Comments 1: The authors manipulated the type of search and the type of target in four experiments. They find that correspondence between the colors of cues preceding the search display decreases performance. There were several interactions involving cue validity, suggesting that effects of correspondence are weaker in singleton detection mode. The authors present an interesting series of experiments. However, there are some errors in the terminology that need to be correct (e.g., “conjunction search”). Also, the results are difficult to follow, and I suggest running a between-experiment ANOVA. The search task is described as conjunction search, but this is not true. In all four experiments, the color and the shape of the target were unique. That is, the target was a green X, but there were no green H or blue X, for example. Thus, it wasn’t necessary to look for a conjunction, but for one of two fixed features. For instance, in Treisman’s studies, there were two feature dimensions (color and shape) and the target was a unique combination of features on the two dimensions (Treisman & Gelade, 1980). For instance, the colors were brown and green, and the shapes were T and X. If the target was a green T, then there would be brown T and a green X. This logic is different from the current experiments. Therefore, the term “conjunction search” is wrong. Rather, there were different combinations of feature and singleton search (according to the terminology in Bacon & Egeth, 1994). |
||
Response 1: Thank you for pointing this out. We agree with the inappropriate terminology of “conjunction search”. Therefore, in the revision, the visual search for the target defined by a combination of feature and category termed as “a hybrid search” rather than a conjunction search. In Line 81, we highlighted and explain the search task “In order to better examine the separate effect of each ACS in a hybrid search task (identifying a target defined by a combination of two attributes, such as a green X), the strength of each ACS can be changed by manipulating different search modes, including feature search mode and singleton search mode (Bacon & Egeth, 1994). ”. Moreover, the “conjunction search” term were modified in the whole manuscript. |
||
Comments 2: The experiments follow a factorial design where feature or singleton search is performed along one or two dimensions. When color is indicated by C and shape by S, whereas feature search is indicated by F and singleton search by S, then the design between experiments is clear: E1: Color: Feature search, Shape: Feature search: C-F, S-F E2: Color: Feature search, Shape: Singleton search, C-F, S-S E3: Color: singleton search, Shape: Feature search, C-S, S-F E4: Color: singleton search, Shape: Singleton search, C-S, S-S This is a very interesting design, and the authors may want to combine the four experiments into a single experiment with four groups. The analysis should be a mixed-factors ANOVA where the between-subject factors 2 (type of shape search: feature, singleton) x 2 (type of color search: feature, singleton) should be included. The within-subject factors should be as before. What are the predictions for this design? In a way, the authors already tested it, but the hypotheses are difficult to extract because they are spread out across four experiments. The new analysis would allow for a much concise manuscript. |
||
Response 2: Thanks for your thorough understanding of our experimental design. As for your advice, we did the mixed ANOVA and reported the results in the revision (Line 427-469). For the hypothesis, we have some predictions in the Introduction. For example, in Line 66-72 : ”To address this confound, the present study defined the target as letters with similar colors (green, including dark green, yellow-green, bright green, cyan, and grass green) and a specific shape ("X"), with shape as the feature level and color as the category level. If the effects of shape-specific ACS (sACS) remain greater than that of color-specific ACS (cACS), it further confirm the different weights of feature- and category-specific ACS. Conversely, it will suggest that the results of Wu et al. (2016) and Wu and Fu (2017) are primarily due to the target-defined properties.” In Line 105-116, “By examining the main effects and interaction effects of SM and CM, the strength of the two types of ACS can be explored: if sACS is stronger than cACS, it is consistent with previous research (Wu et al., 2016; Wu & Fu, 2017; Wu et al., 2013), indicating that the role of ACS is mainly determined by the different processing hierarchies between feature and category, regardless of whether the specific attribute is shape or color. If cACS is stronger than sACS, it indicates that the role of ACS is mainly determined by the shape and color attributes (Kiss et al., 2013), rather than the processing levels of feature and category. Additionally, the homogeneity of distractors was manipulated to change the search mode in four experiments. If different search modes have an impact on the strength of ACS, then the patterns and strengths of sACS and cACS in the four experiments may also differ.” |
||
Comments 3: Figure 2 and following: the bar graphs for the errors are tedious to read. Maybe it would be better to have two separate line plots side by side. One for RTs and the other for errors. |
||
Response 3: Thanks for your comment. We reprinted the graphs with two separate bars side by side, and we combined experimental designs and result diagrams to make the experimental logic clearer. |
||
Comments 4: The authors find that matching colors result in slower RTs than nonmatching colors (main effect of CM, e.g., line 207). Moore and colleagues, however, found that placing an irrelevant color into memory facilitated subsequent responses to the same stimuli (Moore & Weissman, 2011; Moore & Weissman, 2014). The authors should discuss why they find the opposite. That is, worse performance with matching colors. |
||
Response 4: Thanks for your comment. In the revision, we discussed the discrepancy between the current findings and those of Moore and colleagues as follows :”In the present study, the results of main effect of CM showed that matching cues result in slower RTs of target identification relative to nonmatching ones. However, Moore and colleagues found that placing an irrelevant color into memory facilitated subsequent responses to the same stimuli (Moore & Weissman, 2011; Moore & Weissman, 2014). The discrepancy between the findings of current study and those of Moore and colleagues can be attributed to differences in experimental design and cognitive mechanisms involved. The results of Moore and colleagues suggested that memory maintenance can create a cognitive advantage for processing familiar stimuli. In contrast, this study focuses on the immediate effects of color matching on attentional capture during a visual search task. When the cue matches the target color, the cACS is strongly activated, leading to a more intense capture of attention. This intense capture may interfere with other cognitive processes, resulting in slower RTs as the system takes longer to integrate and filter information (Folk et al., 1992; Kiss & Eimer, 2013). The difference in outcomes highlights the complexity of attentional processes, where memory-based facilitation and attentional capture can have opposing effects depending on the task demands and cognitive resources allocated. This study's findings emphasize that in complex visual search tasks, strong attentional capture does not always translate to faster response times, potentially due to increased cognitive load and the need for additional processing to manage the captured attention.” Line 570-588 |
||
Comments 5: I did not quite understand why there were different shades of the target color (green). What was the cue color? The same shade of green? It has been shown that attentional tuning to color is narrower in feature search than in singleton search (Kerzel, 2019), however, it is unclear whether effects of color variation were evaluated at all in the current study. Please clarify. |
||
Response 5: Thanks for your comment. The different shades of greens were used to make the color exist as a category property, that is, multiple colors make up the category of green. The shape and color of cue stimulus were dependent on the degree of shape matching (SM) and color matching (CM) of the current trial. When the cue was C+, its color was one of the five green colors embedded in the “green category”. For the precision of attentional selection, Kerzel (2019) measured effects of cue validity across a range of cue colors from identical up to 60°of separation from the target in CIELAB space. The results found that when the task was made more difficult by increasing the similarity between the target and the nontarget stimuli in the target display, the precision of attentional selection increased, but was still worse than the precision of memory. However, we did not aim to investigate the effects of similarity between cue and target, so that these stimuli in the present study were randomly chosen from a color pool, including red (RGB: 145, 0, 0), blue (RGB: 0, 0, 255), purple (RGB: 123, 0, 123), brown (RGB: 88, 63, 0), and gray (RGB: 69, 69, 69), on the basis of the similar luminance of these colors (13 cd/m2, Spalek et al., 2012). This is a very interesting research topic, and the effects of color variation can be further explored in future studies. We have also added relevant future directions in the revision as follows:”While this study examined the effects of color matching on attentional capture, it did not delve into the precision of attentional tuning to color variation (Kerzel, 2019). Future research could explore how varying levels of color similarity impact the precision and efficiency of attentional selection, providing insights into the mechanisms of attentional tuning in complex visual tasks. ” |
||
Comments 6: line 109: It is not clear to me what is meant by “processing levels of feature and category”. Please explain the terminology |
||
Response 6: Thanks for your comment. The processing of features is occurred at the basic level of cognition while the processing of categories is demanded for higher semantic level. so whether the role of ACS is affected by the processing level (that is, the basic or high level) is said. Sorry for the confusion, we changed this sentence to "the role of ACS is mainly determined by the different processing hierarchies between feature and category,......."(Line 109) |
||
Comments 7: line 116: A justification for the sample size is missing. |
||
Response 7: Thanks for your comment. As for your advice, we added the justification for the sample size in all four experiments as follows:”The sample size calculated by MorePower 6.0 (Campbell & Thompson, 2012, indicated that 34 participants would be required to detect an effect size of Æž2p = 0.2 with a power of 0.80 and α level of 0.05 in an 2×2×2 ANOVA test). Initially,......” |
||
Comments 8: Line 174: If the proportion of valid to invalid cues was 1:1, then the cue predicted the target location. If cue and target location were selected randomly, a ratio of valid to invalid cues of 1:6 would be expected. The authors need to state that the cues were predictive and therefore, not classically “exogenous”. |
||
Response 8: Thanks for your comment. As for your advice, we added the information to Line 179 : “Consequently, the cues were predictive and not classically exogenous. “ Line 173-174 |
||
Comments 9: Line 251: The authors say the effect was significant, but the p-value was .078. The threshold of significance is .05. Therefore, it would be better to write that the effect approached significance. |
||
Response 9: Sorry for the mistake, the 0.078 p value was marginally significant, and we modified it in the revision. We further checked the whole manuscript to make sure there were no similar errors. |
||
Comments 9: Line 254-255: The authors test for differences between valid and invalid for C+ and C-, but there was no interaction between validity and color match. Therefore, there shouldn’t be any post-hoc comparisons. |
||
Response 9: Thanks for your comment. After recalculating the original data, it was found that the initial manuscript was written incorrectly. Instead of the interaction between SM and validity, the edge between CM and validity was marginally significant (Line 259). The simple effect comparison was for this interaction. We modified this mistake and checked the whole data analysis. |
||
Comments 10: Lines 307-310: For the interaction of CM and validity, the authors should test whether the validity effect was significant both for C- and C+. The current t-tests are a mixture of several questions and should be removed. |
||
Response 10: Thanks for your comment. As for your advice, we removed the t-test and added the data analysis :”A spatial cueing effect (higher accuracy in valid condition relative to invalid condition) occurred when the cue was C+, F(1, 39) = 14.89, p < .0 01,η2 = 0.28, but absence when the cue was C- (p = .586), indicating that cACS operate on early attentional orientation.” (Line 320-323) |
||
Comments 11: Line 317-324: The authors find an interaction of CM and validity. The posthoc tests in the text do not match the graphs. The authors write that regardless of the color of the cue, there was a difference between valid and invalid. However, this is not true when looking at Figure 5 where only the C+ conditions result in cueing effects. Please clarify. Similarly, the effect of CM appears reliable for valid trials, but not invalid trials (lines 322-324). Please clarify. |
||
Response 11: Thanks for your comment. After confirming the result again, the drawing is not particularly good, the range of ordinates is too small, and the new drawing will show that there were significant cue effects in both C+ or C- condition. |
||
Comments 12: Line 362: With a p-value of .22, the interaction between SM and CM was not significant. The authors need to revise the sentence saying that it was significant. |
||
Response 12: Thanks for pointing out this mistake, we modified this mistake in the revision. |
||
Comments 13: Lines 392-393: “in the conjunction search task.” This is wrong because the search task was not a conjunction search task. The target was both a shape and a color singleton and the specific combination of shape was not necessary to find it. |
||
Response 13: Thanks for your comment. We agree with the inappropriate terminology of “conjunction search”. Therefore, in the revision, the visual search for the target defined by a combination of feature and category termed as “a hybrid search” rather than a conjunction search. In Line 81, we highlighted and explain the search task “In order to better examine the separate effect of each ACS in a hybrid search task (identifying a target defined by a combination of two attributes, such as a green X), the strength of each ACS can be changed by manipulating different search modes, including feature search mode and singleton search mode (Bacon & Egeth, 1994). ”. Moreover, the “conjunction search” term were modified in the whole manuscript. |
||
3. Response to Comments on the Quality of English Language |
||
4. Additional clarifications |
Reviewer 2 Report
Comments and Suggestions for Authors
The authors investigated the effect of target defining proprieties and search mode on multiple attentional control settings (ACSs) in visual search. It has employed a spatial cueing paradigm (combination of shape and color). Results showed that color specific ACS (cACS) consistently captured attention, influenced by target properties rather than processing hierarchy. Shape-specific ACS (sACS) interacted differently based on experimental conditions. Singleton search modes reduced ACS effects and increased interactions with other factors. The findings suggest that attentional capture depends on target properties and search mode.
This work, which synthesizes the findings from 4 experiments, is meticulously executed and well-written, thus facilitating readers' comprehension of the complex attentional domain. The only suggestion I would express to the authors pertains to the enhancement of the quality of the plots, which, as they are currently presented, is somewhat deficient.
Author Response
Response to Reviewer 2 Comments
|
||
1. Summary |
|
|
Thank you very much for taking the time to review our manuscript. We sincerely appreciate your constructive comments and suggestions. Below, you will find our detailed responses along with the corresponding revisions, highlighted in green in the resubmitted manuscript. We have made every effort to address your concerns thoughtfully, and we value your contribution to improving the quality of this work.
|
||
2. Point-by-point response to Comments and Suggestions for Authors |
||
Comments 1: The authors investigated the effect of target defining proprieties and search mode on multiple attentional control settings (ACSs) in visual search. It has employed a spatial cueing paradigm (combination of shape and color). Results showed that color specific ACS (cACS) consistently captured attention, influenced by target properties rather than processing hierarchy. Shape-specific ACS (sACS) interacted differently based on experimental conditions. Singleton search modes reduced ACS effects and increased interactions with other factors. The findings suggest that attentional capture depends on target properties and search mode. This work, which synthesizes the findings from 4 experiments, is meticulously executed and well-written, thus facilitating readers' comprehension of the complex attentional domain. The only suggestion I would express to the authors pertains to the enhancement of the quality of the plots, which, as they are currently presented, is somewhat deficient. |
||
Response 1: Thanks for your comment. We reprinted the graphs with two separate bars side by side, and we combined experimental designs and result diagrams to make the experimental logic clearer. |
Round 2
Reviewer 1 Report
Comments and Suggestions for Authors
The authors have responded adequately to my earlier criticism. However, the new term selected to describe the search task is unfortunate. “Hybrid search” is mostly used to describe search tasks where participants have to simultaneously search memory and a visual display (e.g., Wolfe, 2012). I suggest using simply “different search tasks with the same target” to refer to the different conditions. The manuscript makes clear which search tasks were used, so there is no need to use a more specific term.
Line 79 : “in a hybrid search task” should be “in different search tasks with the same target letter”
Line 95: “on the search for a hybrid defined target” should be “ for the same target letter in different search tasks”
Line 136: “hybrid search task” by “different search tasks”
Figure 5 is a nice addition to the paper. It could be further improved if the labels on the y-axis included the search mode (Shape: feature + Color: singleton, etc.). This would make it easier for the reader to compare the results between experiments (without having to memorize all the information).
Wolfe, J. M. (2012). Saved by a log: how do humans perform hybrid visual and memory search? Psychological Science, 23(7), 698-703. https://doi.org/10.1177/0956797612443968
Author Response
Response to Reviewer 1 Comments
|
||
1. Summary |
|
|
Thank you very much for taking the time to review our manuscript. We sincerely appreciate your constructive comments and suggestions. Below, you will find our detailed responses along with the corresponding revisions.
|
||
2. Point-by-point response to Comments and Suggestions for Authors |
||
Comments 1: The authors have responded adequately to my earlier criticism. However, the new term selected to describe the search task is unfortunate. “Hybrid search” is mostly used to describe search tasks where participants have to simultaneously search memory and a visual display (e.g., Wolfe, 2012). I suggest using simply “different search tasks with the same target” to refer to the different conditions. The manuscript makes clear which search tasks were used, so there is no need to use a more specific term. Line 79 : “in a hybrid search task” should be “in different search tasks with the same target letter” Line 95: “on the search for a hybrid defined target” should be “ for the same target letter in different search tasks” Line 136: “hybrid search task” by “different search tasks” |
||
Response 1: Thank you for your comment. We agree with the inappropriate terminology of “hybrid search”. As for your advice, in the revision, we simply used“different search tasks with the same target” to refer to the different conditions. Thanks for your kindly suggestion, we revised“in a hybrid search task”to“in different search tasks with the same target letter”, “on the search for a hybrid defined target” to “ for the same target letter in different search tasks”, and “hybrid search task” to “different search tasks” And we checked the whole manuscript to avoid the terminology of “hybrid search”. |
||
Comments 2: Figure 5 is a nice addition to the paper. It could be further improved if the labels on the y-axis included the search mode (Shape: feature + Color: singleton, etc.). This would make it easier for the reader to compare the results between experiments (without having to memorize all the information). Wolfe, J. M. (2012). Saved by a log: how do humans perform hybrid visual and memory search? Psychological Science, 23(7), 698-703. https://doi.org/10.1177/0956797612443968 |
||
Response 2: Thanks for your comment. In the revision, We have labeled y-axis by marking the search mode in each Experiments. |